# Huntingtin fibrils with different toxicity, structure, and seeding potential can be interconverted

J. Mario Isas[1], Nitin K. Pandey[1], Hui Xu[1], Kazuki Teranishi[1], Alan K. Okada [1,2], Ellisa K. Fultz[1], Anoop Rawat[1], Anise Applebaum[1], Franziska Meier[1], Jeannie Chen [1], Ralf Langen [1✉] & Ansgar B. Siemer [1✉]

The first exon of the huntingtin protein (HTTex1) important in Huntington's disease (HD) can form cross-β fibrils of varying toxicity. We find that the difference between these fibrils is the degree of entanglement and dynamics of the C-terminal proline-rich domain (PRD) in a mechanism analogous to polyproline film formation. In contrast to fibril strains found for other cross-β fibrils, these HTTex1 fibril types can be interconverted. This is because the structure of their polyQ fibril core remains unchanged. Further, we find that more toxic fibrils of low entanglement have higher affinities for protein interactors and are more effective seeds for recombinant HTTex1 and HTTex1 in cells. Together these data show how the structure of a framing sequence at the surface of a fibril can modulate seeding, protein-protein interactions, and thereby toxicity in neurodegenerative disease.

[1] Department of Physiology & Neuroscience, Zilkha Neurogenetic Institute, Keck School of Medicine, University of Southern California, Los Angeles, CA, USA. [2] Present address: Department of Emergency Medicine, Regions Hospital, St. Paul, MN, USA. ✉email: langen@usc.edu; asiemer@usc.edu

Huntington's disease (HD) is a debilitating neurodegenerative disease caused by mutant huntingtin with an expanded polyQ region containing more than 35 consecutive Gln residues. Such polyQ expansions render the huntingtin protein and its biologically occurring N-terminal fragments more aggregation prone[1]. Formation and deposition of aggregated, amyloid-like material from N-terminal fragments of huntingtin is one of the hallmarks of HD. As in other amyloid diseases, a variety of misfolded and aggregated species are formed, but not all of these species contribute equally to toxicity. While some species may be toxic, other species may be less toxic and potentially even protective. This difference is likely to account for findings on the various amyloidogenic proteins, where varying degrees of toxicity have been reported in biological settings, or where different pathways of toxicity appear to be triggered by misfolding. To characterize the roles played by different misfolded species and to foster diagnostic and therapeutic approaches, it is important to understand the structural and functional properties of the various misfolded forms of huntingtin.

Huntingtin exon-1 (HTTex1) is one of the biologically most important N-terminal huntingtin fragments[2]. HTTex1 contains the disease causing polyQ domain, which is flanked by the N-terminal N17 domain and a C-terminal, proline-rich domain (PRD). Mutant HTTex1 and fragments of similar size are found in diseased tissue and are naturally generated by proteolysis or aberrant splicing[3]. The importance of mutant HTTex1 in HD is further underscored by animal model studies, which show that expression of mutant HTTex1 is sufficient for causing disease symptoms[4]. Like many other amyloidogenic proteins, mutant HTTex1 forms polymorphic fibrils. In vitro, the yield of a given polymorph is governed by several factors, including temperature[5,6]. It has previously been reported that HTTex1 fibrils grown at 4 °C are more toxic than those grown at 37 °C[5]. Both fibril types appear to be present in vivo, but the affected brain regions have been reported to have a higher proportion of the more toxic fibril types[5]. We have previously investigated the structural features of the more toxic fibril type by electron paramagnetic resonance (EPR) and solid-state nuclear magnetic resonance (NMR). This work has led us to propose the bottlebrush model in which the polyQ region forms the β-sheet core region, while the C-terminal PRD projects outward like the bristles in a bottlebrush[7,8]. Cryo-electron microscopy of HTTex1 in cellular inclusions supported this model[9]. The N17 was found to be ordered and in agreement with α-helical structure determined by solid-state NMR[10]. Different circular dichroism (CD) and infrared spectral features suggested structural differences between the fibril types of different toxicity[5], but in contrast to the structural information available for the fibrils grown at 4 °C and room temperature,[6–8,10–13], little is known about the less toxic fibrils grown at 37 °C. Knowing the structural features that modulate toxicity could be important for structure-based therapeutic strategies. In addition to a structural characterization, a number of other questions remain unanswered. It is not known whether the different fibril types represent distinctively different strains as seen ubiquitously for other amyloid proteins. Moreover, the seeding behavior of the different fibril types has not been investigated. Seeding plays an important role in misfolding and aggregate formation and it is likely that several types of misfolded proteins can spread within the brain using a seeding mechanism. Thus, in principle different seeding behaviors could have significant impact on aggregate toxicity. In the present study, we combined EPR, solid-state NMR, and other biophysical methods to determine the structural differences between fibrils grown at 37 and 4 °C and to test whether they have different seeding abilities in the test tube and in cells.

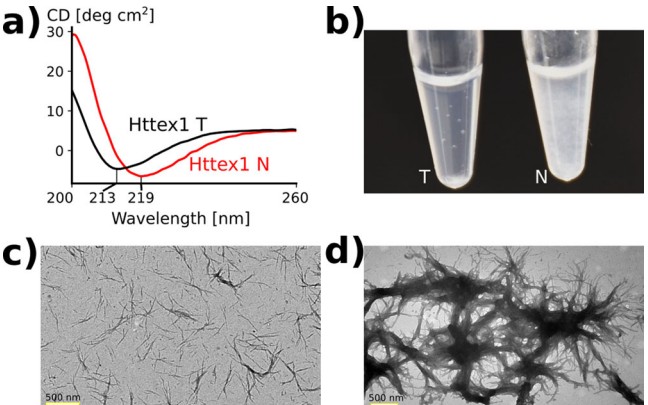

**Fig. 1 HTTex1 T and N-fibril types have distinct CD minima, visible appearance, and EM signatures. a** CD spectra of T- and N-fibrils are plotted in black and red, respectively. There is a distinct shift in the minimum of the spectrum as reported previously[5]. **b** Visual appearance of both fibrils in suspension. The T-fibrils appear translucent whereas the N-fibrils have a cloudy appearance. **c** The EM image of T-fibrils shows little bundling. **d** EM image of N-fibrils shows predominantly bundled fibrils. These observations are highly reproducible and have been repeated dozen of times.

## Results

**Growth conditions determine fibril type of HTTex1.** It has previously been reported that HTTex1 fibrils at two different temperatures, 4 or 37 °C, results in the formation of toxic or nontoxic fibrils, respectively[5]. In order to characterize the structural differences between those fibril types, we used the established conditions to generate the previously described toxic and nontoxic fibrils from HTTex1(Q46). For simplicity, we refer to these fibrils as T- and N-fibrils, respectively. In agreement with the original study, we found that the two different fibril types exhibited distinctively different CD spectra (Fig. 1a). While T-fibrils typically had a minimum around 213 nm, that of N-fibrils was significantly shifted over to higher wavelengths (218–225 nm). These data are indicative of significant structural differences between the different fibril types. An additional, difference between the two fibril types was their optical appearance. While the T-fibrils were translucent, the N-fibrils were turbid, indicating that these fibrils were larger with a higher propensity to scatter light (Fig. 1b), which likely caused their lower relative CD intensity (Supplementary Fig. 1). Consistent with the optical appearance, EM images showed that T-fibrils were mostly non-bundled with typical diameters of 12 ± 2 nm (Fig. 1c) while the N-fibrils were much thicker (220 nm to 10-μm diameter) with a more bundled appearance (Fig. 1d).

**N-fibrils have a less flexible PRD compared to T-fibrils.** We next used a combination of continuous wave (CW) and pulsed EPR as well as solid-state NMR to determine in which regions the structural changes might lie. For the CW EPR experiments, several spin-labeled derivatives were generated harboring singly labeled sites within the N17, polyQ, or PRD (Fig. 2a). T- or N-type fibrils were then grown from these derivatives and the corresponding EPR spectra were recorded. To ensure that proper fibril types were generated, each preparation was tested using CD (Supplementary Fig. 1). The EPR spectra of T-fibrils for sites in the N17 (15R1, 17R1; Fig. 2b, black spectra) and the polyQ region (30R1, 48R1; Fig. 2b) revealed multicomponent EPR spectra that were dominated by highly immobilized components. In contrast, the sharp and narrowly spaced spectral lines for position 63 (last residue in polyQ) and the sites in the PRD (71R1, 76R1, 86R1)

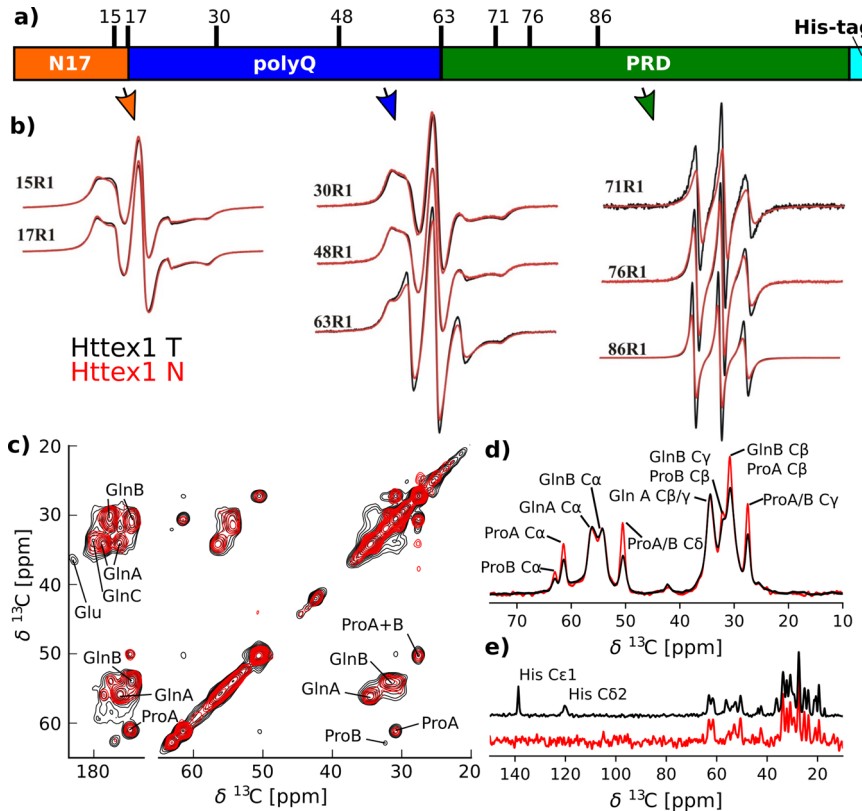

**Fig. 2 EPR and solid-state NMR spectra indicate a more dynamic PRD in the T-fibrils compared to the N-fibrils. a** Domain structure of HTTex1 highlighting the N17 region in orange, the polyQ domain in blue, the PRD in green, and the C-terminal His-tag in cyan. The residues that were spin-labeled for CW measurements are indicated. **b** CW EPR spectra of T- and N-fibrils are shown in black and red, respectively. The residue number of the spin label is indicated. The spectra of N- and T-fibrils overlap very well for all positions within the N17 and polyQ region indicating that they have the same structure and dynamics. However, the spectra of the T-fibrils are narrower for the last Q in the polyQ region (63R1) and all sites within the PRD indicating increased dynamics in this region compared to N-fibrils. **c** The 2D DARR $^{13}$C-$^{13}$C spectra of N- and T-fibrils overlap very well, indicating that the static fibril core of both fibrils has a very similar structure. The spectra of N- and T-fibrils are shown in red and black, respectively. Both spectra were recorded at 0 °C, 25-kHz MAS with 50-ms mixing time. The spectrum of the T-fibrils was previously published as Fig. 2 by Isas et al.[8]. **d** 1D $^{13}$C cross polarization (CP) spectra, which are sensitive to more static residues. Spectra of N-fibrils (red) and T-fibrils (black) were normalized to their Gln Cα intensities. The amino acid assignments of the lines are indicated. The higher intensities of the proline lines in the spectra of the N-fibrils indicate that this domain is more static in this fibril type. **e** Refocused INEPT spectra, which detect highly dynamic residues, were normalized according to the spectra shown in (**d**). The assignment of the His resonances that were detected for the T but not for the N-fibrils is indicated.

indicated much higher mobility in this region. These results were very similar to those previously obtained for fibrils grown at low temperature under similar, but not identical conditions[7]. Together with our earlier studies, they support the bottlebrush model where the PRD bristles face away from the central core of the fibril[8].

When we compared the EPR spectra from T-fibrils to those from N-fibrils (red spectra in Fig. 2b), we found them to be nearly identical for sites in the N17 and polyQ indicating that the local structure in these regions is highly related in the two different fibril types. Subtle differences, however, were seen at position 63 and these differences became more noticeable for sites in the PRD region. For each of these labeling sites, the spin-normalized EPR spectra of the T-fibrils had higher amplitudes than those of N-fibrils. This was caused by the more pronounced presence of sharp lines in the T-fibril spectra that indicate high mobility. Based upon these data, the T- and N-fibrils have related structural features in the N17 and polyQ, but the N-fibrils have more strongly immobilized PRD regions.

To further investigate the differences between the different fibril types, we measured solid-state NMR spectra of T- and N-fibrils. To compare the fibril core of these spectra, we used cross polarization (CP) based solid-state NMR spectra that rely on

dipolar couplings and that primarily detect the static parts of an amyloid fibril. An overlay of $^{13}$C-$^{13}$C 2D DARR spectra of T- (black) and N-fibrils (red) is shown in Fig. 2c. Both spectra were remarkably similar. Each spectrum contained the two Gln peaks (Gln A and Gln B) that had previously been shown to be the β-sheet structure formed by the polyQ domain[8,10,13]. The difference between Gln A and Gln B have been hypothesized to be the result of alternative side chain conformations[12,13]. In addition, cross peaks corresponding to Pro in a polyP II (PPII) conformation (ProA) and weak cross peaks corresponding to Pro in a random coil conformation (Pro B) could be detected in both spectra. All of these peaks superimposed very well in both spectra indicating a highly similar structure. The overall conclusion that N- and T-fibril cores have a similar overall structure is also consistent with the 1D CP spectra shown in Fig. 2d, as T- and N-fibrils give rise to nearly identical peak positions. However, more detailed analysis of the peak intensities reveals that the Pro peaks are comparatively smaller in the T-fibrils. Inasmuch as the CP spectra are more sensitive to static regions, this indicates that the T-fibrils have a smaller proportion of their Pro residues in a static state.

While CP-based spectra focus on the static parts of the fibrils, INEPT-based spectra of fully protonated protein at moderate

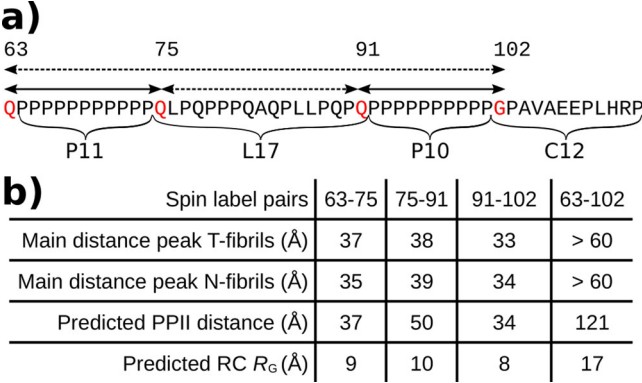

**a)**

**b)**

| Spin label pairs | 63-75 | 75-91 | 91-102 | 63-102 |
|---|---|---|---|---|
| Main distance peak T-fibrils (Å) | 37 | 38 | 33 | > 60 |
| Main distance peak N-fibrils (Å) | 35 | 39 | 34 | > 60 |
| Predicted PPII distance (Å) | 37 | 50 | 34 | 121 |
| Predicted RC $R_G$ (Å) | 9 | 10 | 8 | 17 |

**Fig. 3 Long-range distance inside the PRD of N- and T-fibrils is highly similar. a** Sequence of the PRD indicating the positions of the labels introduced for distance measurements in red and the distance pairs as arrows. The segments of the PRD including the two polyP regions (P11 and P10) and the linker region (L17) and C-terminals region (C12) are also indicated. **b** Table listing all distances measured via EPR DEER experiment for each spin label pair and fibril type. The distance corresponds to the maximum of the main peak derived from the DEER decay curve (DEER data and fits are shown in Supplementary Fig. 2). In addition, the theoretical distances for a polyP II helix assuming an increase of 3.1 Å per residue and the theoretical radius of gyration ($R_G$) for a random coil, are given ($R_G = R_O N^\nu$ with $R_O = 1.927$ Å, $\nu = 0.598$, and $N$ is the number of residues). Both distances spanning the polyP regions (P11 and P10) correspond nicely to a PPII distance. The distance spanning the L17 region is significantly shorter than a PPII helix and longer than expected for a random coil structure.

magic angle spinning (MAS) frequencies are selective for highly dynamic domains[14–16]. The 1D $^{13}$C refocused INEPT spectra (Fig. 2e) show that both T- and N-fibrils have considerable dynamic domains. However, a hallmark of the T-fibrils is the presence of His $C\varepsilon_1$ and $C\delta_2$ resonances that are not detected in N-fibrils. This difference indicates that the C-terminal His-tag is more dynamic in T-fibrils compared to N-fibrils. Together the 1D CP and 1D INEPT data are consistent with the increased mobility of the PRD observed for T-fibrils by EPR (Fig. 2b).

Having obtained EPR and NMR-based evidence that the main differences between the two fibril types reside in the PRD, we next investigated this region structurally by pulsed EPR (DEER)-based distance measurements. Toward this end, we generated four different labeling pairs in the PRD. Two of the labeling pairs, 63–75 and 91–102, flank the two uninterrupted polyP repeats P11 and P10, respectively. The 75–91 pair borders a Pro-rich linker region L17, which connects the two polyP repeats. The 63–102 labeling pair was designed to obtain distance information encompassing the entire PRD (Fig. 3a). We sparsely incorporated these derivatives (10–5%) into N or T-fibrils grown from excess (90–95%) unlabeled protein. This dilution of spin-labeled monomers with a majority of unlabeled protein avoids the influence of intermolecular distances in our measurements, which is further removed via background subtraction. The very slow decay of the DEER signal for the 63–102 labeling pair (Supplementary Fig. 2) indicates the absence of short intermolecular distances and supports the success of this dilution approach.

We first investigated the distances of the T-fibrils. As expected for regions that contain large amounts of PPII helices[17], the distance distributions for the 63–75 and 91–102 labeling pairs are broad (Supplementary Fig. 2). As summarized in Fig. 3b, the peaks of the distance distributions (37 and 33 Å, respectively) correspond remarkably well to the predominant, fully extended form of a PPII helix, which increases in length by ~3.1 Å/amino

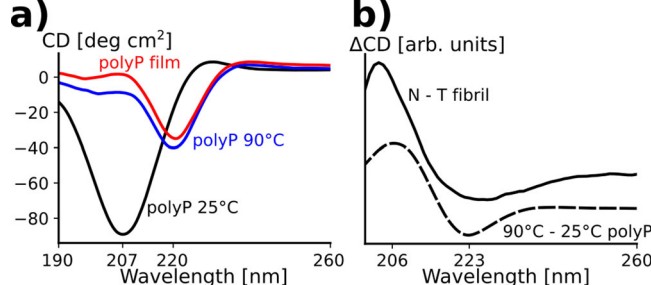

**Fig. 4 PolyP and HTTex1 difference spectra suggest an increased entanglement of the PRDs in N compared to T-fibrils. a** CD spectra of polyPro at a concentration of 0.1 mg/ml recorded at 25 °C (black) and 90 °C (blue). The spectrum in red was recorded after the spectrum at 90 °C and after removing the liquid from the CD cuvette indicating the formation of a polyP film inside the cuvette. **b** Difference of N- and T-fibril CD spectra of Fig. 1 (solid line) and polyP CD spectra recorded at 90 and 25 °C (dashed line). The shared maxima at about 206 nm and minima at about 223 nm indicate that the structural differences that occur during heat-induced polyP film formation are comparable to those between HTTex1 N and T-fibrils.

acid (37 and 34 Å, respectively). These distances are not consistent with a random coil structure that would, from theoretical considerations, result in much shorter distances[18]. The distance for the 75–91 labeling pair (38 Å) was also much longer than expected for random coil structure, but it was also significantly shorter than expected for a perfect PPII helix (50 Å). Together with the solid-state NMR data, which revealed a mixture of PPII helical and random coil structure for proline residues in the PRD[8,14,19], this indicates the formation of a less than fully extended PPII helical structure between residues 75–91. Despite this imperfect PPII helical structure between residues 75 and 91, we found no evidence for a folding back of the PRD onto itself, as the distance for the 63–102 pair was very long, beyond the limit of detection (~60 Å), indicating that residues 75–91 introduce only a slight kink. Interestingly, highly similar distance distributions for the respective labeling pairs were also observed in N-fibrils, indicating that overall conformations of the PRD bristles are similar. Collectively, the solid-state NMR, CW, and DEER EPR data indicated that the overall structure in the PRD region was similar in N- and T-fibrils with significant amounts of PPII helical structure present in both cases. The central difference between the two fibril types, however, resides in the degree of packing interactions in the PRD bristles. While these bristles freely radiate outward in T-fibrils as shown previously[7,8], they are much more tightly entangled in the thicker and more bundled N-fibrils. Considering that the two fibril types are structurally highly similar except for the thickness of the fibrils and the packing of the bristles, it may be possible that N-fibrils are made up from T-fibril building blocks that are held together by interactions between their PRD bristles.

**Bundling of N-fibrils is driven by polyP film formation**. To further test the notion that proline-rich regions could mediate protein–protein interactions, we investigated the oligomerization properties of polyP peptides using CD experiments. At low temperatures, the polyP peptides exhibited a characteristic CD signature for a PPII helix with a minimum at 207 nm and a small, but detectable maximum near 230 nm (Fig. 4a, black line). Upon heating, an entirely different spectrum with a minimum near 220 nm appeared (Fig. 4a, blue line). Similar transitions have been reported and interpreted in terms of temperature-dependent oligomerization or film formation of the polyP peptides[20]. To test

for film formation in our polyP preparations, we removed the liquid from the CD cuvette at high temperature and recorded the CD spectra (Fig. 4a, red line). Much of CD spectrum remained after liquid removal indicating that the peptides no longer resided in solution but rather formed a film on the surface of the glass[20]. Interestingly, polyP film formation causes a red shift in the minimum of its CD spectrum that is reminiscent of the difference between T- and N-fibrils. To investigate these potential similarities further, we generated the difference spectra for the polyP peptides at high and low temperature as well as T- and N-fibrils (Fig. 4b). The respective difference spectra have overall similar shapes with maxima near 206 nm and minima near 223 nm. These results are consistent with the notion that the entangeling of PRD bristles could be linked to the generation of N-fibrils.

**T- and N-fibrils can interconvert.** The overall similarity between T- and N-fibrils and the propensity of polyP region to self-aggregate suggested that N-fibrils might arise from T-fibrils via entanglement of PRD bristles. To investigate this hypothesis further, we generated T-fibrils and followed their long-term stability over time. As shown in Fig. 5a, the CD spectra slowly changed over time with the minimum shifting to larger wavelength, eventually approaching the CD spectrum of N-fibrils after 9 days at 4 °C. The change in the CD spectra coincided with reduced amounts of the sharp (mobile) component in the EPR spectra of 81R1 (Fig. 5b) as expected for a transition to N-fibrils. We also re-measured the 1D $^{13}$C refocused INEPT solid-state NMR spectrum on the same sample of T-fibrils after 12 months and found that the of His $C\varepsilon_1$ and $C\delta_2$ resonances became undetectable similar to the corresponding spectrum of N-fibrils (Fig. 5c).

Having established that T-fibrils can slowly transition into more entangled, N-like fibrils, we were wondering whether the transition from T- to N-fibrils could be reversed. We found that treatment of N-fibrils with 0.5% trifluoroacetic acid (TFA) in $H_2O$ yielded a strong blue shift in the CD spectra of N-fibrils (Fig. 6a), a transition characteristic for the formation of T-fibrils (Fig. 1a). These data further supported that N-fibrils are formed from T-fibrils through the entanglement of the PRD region, which can be reversed by treatment with organic solvent. Interestingly, we noticed that treatment of N-fibrils with hexafluoroisopropanol (HFIP) followed by 0.5% TFA resulted in progressively blue shifted minima in the CD spectra (Fig. 6a). This change was accompanied by the formation of increasingly disaggregated fibrils (Fig. 6c). Thus, this treatment disaggregated N-fibrils into fibrillar structures that were shorter and even less bundled than T-fibrils. Inasmuch as we also saw the formation of such short fibrils during early time points of T-fibril formation, we refer to them as protofibrils or P-fibrils. If N-fibrils can be successively converted into T- and P-fibrils, one would expect that directly applying the HFIP/TFA treatment to T-fibrils should result in P-fibrils. Indeed, this is the case as shown in Fig. 6d. In addition, P-fibrils can also be made from T-fibrils using 0.5% TFA and sonication (Supplementary Fig. 3). P-fibrils made that way were used in our toxicity assays described below.

In order to structurally characterize the P-fibrils, we used solid-state NMR. Figure 6e shows the comparison of 1D $^{13}$C refocused INEPT and CP spectra of HTTex1 T- and P-fibrils. The CP spectra, which were normalized to the intensity of the Gln $C\alpha$ peaks, overlap perfectly indicating that the structure of the static fibril core of these two fibril types is very similar. This is confirmed by the overlap of the 2D DARR spectra shown in Fig. 6f. However, the intensities of the refocused INEPT spectra, which were scaled according to the CP spectra, are different. The higher overall signal observed for the P-fibrils indicates that the

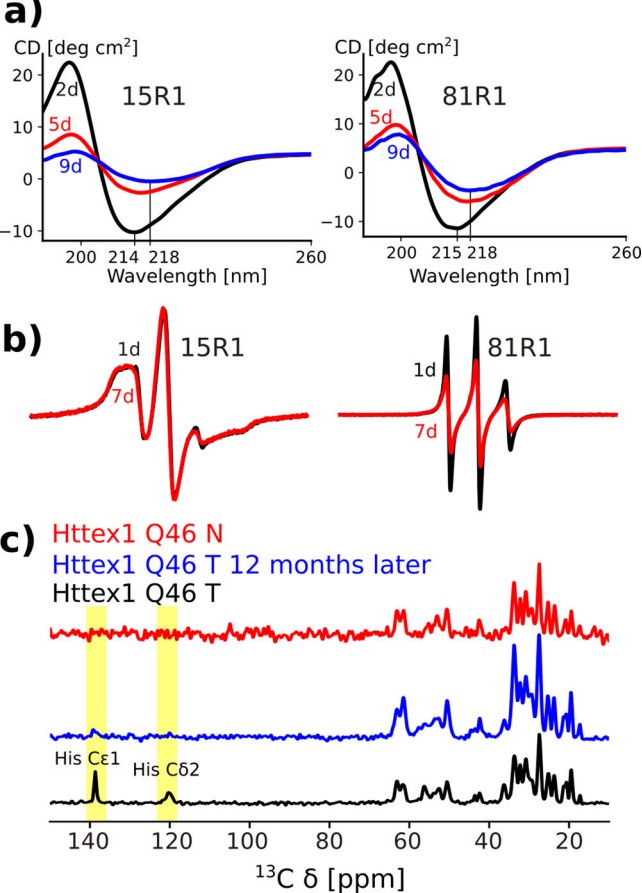

**Fig. 5 Over time T-fibrils turn into N-fibrils. a** CD spectra of HTTex1(Q46) fibrils that were spin-labeled at the N-terminus (15R1) and the PRD region (81R1). Spectra of fibrils incubated at 4 °C for 2 days (black), 5 days (red), and 9 days (blue) are shown. The decrease in apparent CD and the blue shift of the minima are compatible with a transition from T- to N-fibrils. **b** CW EPR spectra of corresponding fibril preparations after 1 day (black) and 7 days of incubaiton at 4 °C (red). The spectrum of the N-terminal 15R1 stays unchanged, whereas the spectrum of 81R1 decreases in intensity over time indicating a decrease in mobility. **c** Refocused INEPT spectra of HTTex1(Q46) T-fibrils (black), the same fibrils 12 months after fibrillization (blue), and N-fibrils (red). The His resonances, which are detected in freshly prepared T-fibrils, disappear after 12 months and are neither visible in N-fibrils.

C-terminal domain in these fibrils is more dynamic than in the T-fibrils. Except for the shift of the His $C\varepsilon_1$ line seen in the refocused INEPT spectrum, no chemical shift change was observed between the N-, T-, and P-fibrils (see also comparison of INEPT HETCOR spectra in Supplementary Fig. 4).

**Bundling reduces antibody accessibility of PDR.** Do the fibril types described above show a different degree of interaction to known HTTex1 fibril binding partners? To answer this question, we used dot blots with the MW8 antibody and an anti-polyHistidine antibody as control. MW8 recognizes an epitope in the PRD and is known to exclusively bind to HTTex1 fibrils[21,22]. Removing this epitope by replacing the Pro-rich linker in the PRD with prolines abolishes the binding of MW8 to HTTex1[23]. As can be seen from Supplementary Fig. 5, MW8 binds strongly to P- and T-fibrils, less efficient to N-fibrils, and does not bind to the soluble HTTex1 fusion protein or the

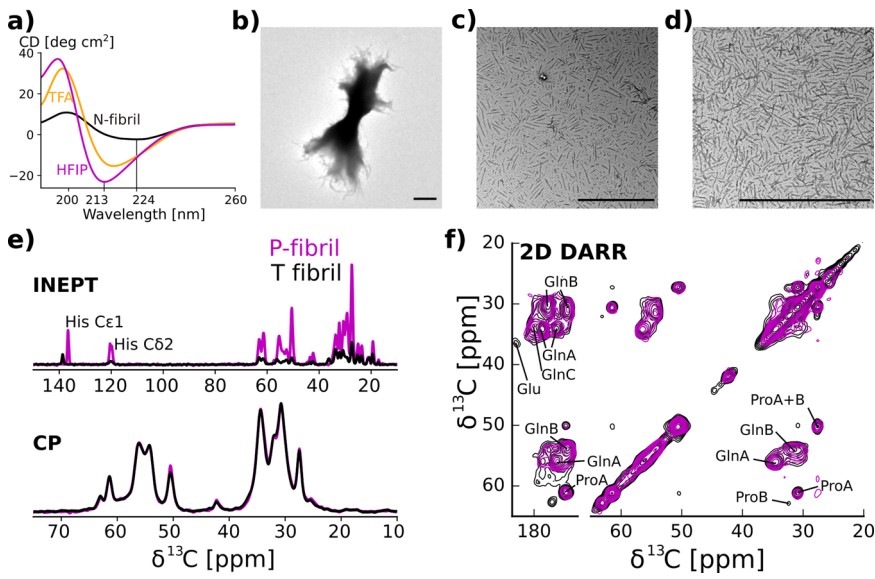

**Fig. 6 TFA and HFIP can disaggregate T-fibrils to form highly disentangled P-fibrils. a** CD spectra of N-fibrils (black), N-fibrils that were treated with 0.5% TFA in $H_2O$ (orange), and HFIP (purple). The minima of the spectra treated with 0.5% TFA and HFIP are increasingly blue shifted and show an increase in apparent circular dichroims likely due to a decrease in scattering. **b** EM of highly bundled N-fibrils before HFIP treatment. **c** EM of same fibrils after HFIP treatment showing the presence of highly unbundled protofibrils. **d** EM of T-fibrils after HFIP/TFA treatment. Scale bars for (**b**)–(**d**) denote 1 µm. **e** 1D $^{13}C$ spectra of HTTex1(Q46) P- and N-fibrils are shown in violet and black, respectively. CP spectra were normalized to their Gln $C_\alpha$ intensities. The close to perfect overlap of the CP spectra indicates that the static core of these two fibril types is very similar. Refocused INEPT spectra were normalized according to the CP spectra. His side chain carbon resonances are indicated. The much higher relative intensity of the INEPT spectra of the P-fibrils indicates that the PRD is more dynamic. The shift in the His $C_{\varepsilon 1}$ line reflects the low pH used for the protofibril preparation. **f** Overlap of 2D DARR $^{13}C$-$^{13}C$ spectra of HTTex1 (Q46) P-fibrils (violet) and T-fibrils (black) indicates that the static fibril core of both fibrils has a very similar structure. Both spectra were recorded at 0 °C, 25 kHz MAS with 50 ms mixing time. The spectrum of the 4 °C fibrils was previously published as Fig. 2 by Isas et al.[8].

HTTex1 mutant lacking the MW8 epitope (HTTex1 Pro). Interestingly, dissociating N-fibrils via sonication similar to the production of HTTex1 seeds, increase MW8 binding. Together, these data indicate that less entangled and more dynamic PRDs allow the fibril to engage in more protein–protein interactions.

**T- and P-fibrils are more seeding competent than N-fibrils.** Seeding has long been considered an important aspect by which the misfolding of amyloidogenic proteins is rapidly propagated. We therefore tested the differential ability of P-, T-, and N-fibrils to act as seeds. We performed seeding reactions using recombinant HTTex1 and used the previously developed time-dependent change in EPR signal of HTTex1 labeled at position 35 as a readout[24]. As shown in Fig.7a, b, all fibril types enhanced the loss in signal amplitude, indicating that they all were able to act as seeds. The seeding efficiency was most pronounced in the case of P-fibrils, while that of T-fibrils was weaker and that of N-fibrils was weakest.

To confirm these results in the context of cells, we transfected Neuro2a cells with HTTex1-EGFP harboring different Q-lengths[25] and tested how exogenously added fibrils promote aggregate formation of endogenous HTTex1-EGFP within the cells. As shown in Fig.7c, d, P-fibrils were most potent at seeding as revealed by the strongly enhanced propensity to form puncta for cells expressing HTTex1-EGFP Q72 or Q39. This effect was stronger than for the T-fibrils, which, nonetheless, still exhibited strong seeding ability. In contrast, N-fibrils had a poor seeding ability as puncta formation for these fibrils was comparable to that of the unseeded case. As controls, we also used cells expressing HTTex1-EGFP with a short Q-length (Q16), which are known to have a low fibril forming propensity or EGFP alone. Neither of these cells exhibited any detectable change in puncta from treatment with the different fibrils indicating that seeding

effect, which was quantified by the relative puncta formation of all transfected cells, is specific to HTTex1 with expanded Q-lengths. We did not observe any significant change in transfection efficiency for all HTTex1-EGFP constructs independent of Q-length and seeding.

**P- and T-fibrils are toxic.** In addition, we estimated the toxicity of our fibril types by adding them exogenously to STHdh Q7/111 cells, which are derived from the striatal neurons from heterozygous $Hdh^{Q111}$ knock-in mouse embryos[26] and determining the resulting cell death via nuclear fragmentation. As can be seen from Fig. 7e, f, DAPI-labeled cells showed a significantly larger amount of nuclear fragmentation (arrows in Fig. 7e) for T- and P-fibrils, whereas N-fibrils showed as much cell toxicity as our buffer control. In summary, our data show that less entangled PRDs result in fibrils that can seed soluble HTTex1 much more potently. The ability to propagate more efficiently is another important aspect that help explain the higher toxicity of T- and P-fibrils.

**Discussion**

Using a combination of CW and pulsed EPR, solid-state NMR, CD, and EM, we investigated the structural features of three different fibril types, two of which were previously found to have very different toxicities. Despite the different CD spectra of the fibril types, the overall architecture of their respective core regions was remarkably similar according to CW EPR and solid-state NMR. While the structures of the individual PRD bristles were also very similar, their dynamics varied among the different fibril types. T- and P-fibrils exhibited the characteristic bottlebrush structures where bristles have high mobility and radiate outward. In contrast, the bristles in N-fibrils are less dynamic and coalesced resulting in highly bundled fibrils (Fig. 8). That the bundling of

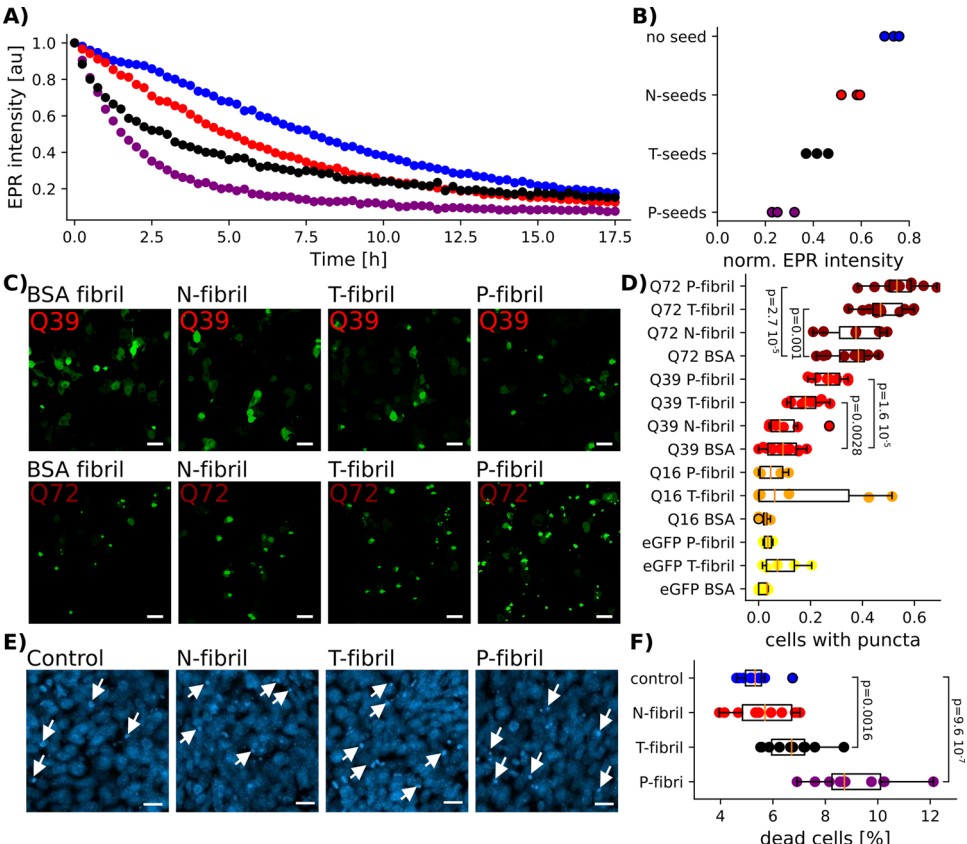

**Fig. 7 HTTex1 fibrils with disentangled PRDs are more efficient in seeding soluble HTTex1. a** Time-dependent CW EPR intensities of unseeded HTTex1 (Q46) 35R1 (blue) and HTTex1(Q46) 35R1 seeded with N- (red), T- (black), and P-fibrils (purple). **b** EPR intensity of HTTex1(Q46) 35R1 4 h after initiating fibril formation of three separate experiments ($n = 3$). The decrease in intensity, which is a reporter of fibril formation, is fastest when seeding with P-fibrils followed by seeding with T-, N-fibrils, and unseeded protein. **c** Puncta formation in Neuro2a cells transfected with HTTex1 with varying polyQ length, and a C-terminal EGFP tag and EGFP as control. Cells were co-transfected with N-, T-, and P-fibril seeds and BSA fibril seeds as control. Fluorescence microscopy images were taken 48 h after transfection. Representative images for HTTex1Q39 (red) and Q72 (burgundy) are shown. Scale bars denotes 40 µm. Puncta formation was measured in 3–11 biological replicates and was reproducible as illustrated in (**d**). **d** Fractions of transfected cells containing puncta depending on HTTex1 polyQ length and nature of co-transfected fibrils seeds. Fractions of separate experiments are reported together with boxplots that show 25th to 75th percentiles, the median in orange, and whiskers from minimum to maximum. Significant *P* values from a two-sided *t*-test are indicated. P-fibril seeds led to the strongest induction of puncta in cells transfected with HTTex1Q72 (burgundy) and Q39 (red) followed by T-fibrils. The effect of N-fibril and BSA fibril seeds on puncta formation was roughly the same. Few puncta were observed for HTTex1Q16 (orange) and EGFP (yellow) independent of co-transfected fibril seeds. **e** Fibril toxicity using ST*HDH Q7/111* cells. Representative images of DAPI-labeled cells to which fibril types were exogenously added. White arrows point DAPI-labeled brighter small puncta, which are recognized as dead cells. Scale bars denote 20 µm. Data were reproducible as illustrated in (**f**). **f** Percent of dead cells detected in ten images coming from three biological replicates. Each individual percentage together with a boxplot is shown. Boxplots indicate 25th to 75th percentiles, the median in orange, and whiskers from minimum to maximum. Significant *P* values from two tailed *t*-tests are indicated.

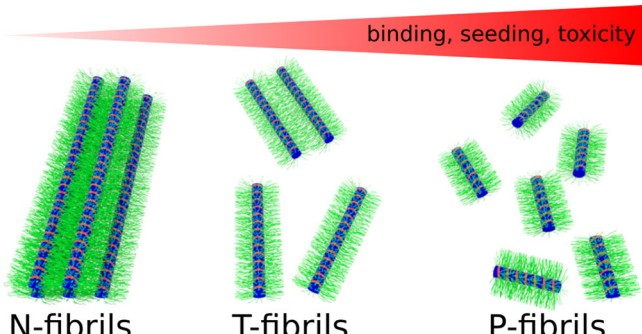

**Fig. 8 Model of the N-, T-, and P-fibrils.** The PRDs of the N-fibrils (green) are entangled resulting in bundling and consequently less accessible fibril surfaces. The PRDs of the T-fibrils are keeping the individual fibrils more separate. The P-fibrils are shorter and their PRD are even less restricted and more accessible.

N-fibrils is caused by PRD aggregation in a mechanism analogous to polyPro film formation is supported by the following experimental findings. (I) EPR data showing only a change in the PRD with bundling. (II) Solid-state NMR data showing a less dynamic PRD in N compared to the T- and P-fibrils. (III) CD spectra showing that the structural difference between bundled and unbundled fibrils is analogous to polyP film formation. (IV) A dot blot showing that the antibody MW8, which recognizes an epitope in the PRD, preferentially binds T- and P-fibrils. In addition, our EPR, solid-state NMR, and CD data support the notion that N-fibrils can form from T-fibrils via the coalescence of the PRD bristles. Our model is furthermore consistent with the greater diameter of N-fibrils and the fact that all fibril types can be interconverted.

The finding that HTTex1 can exist in a variety of different fibril types is a property that is shared with the vast array of amyloidogenic proteins for which fibril polymorphs have been described[27–32]. What is unusual about the HTTex1 fibril types is

that they are different from strains that have been observed in the other amyloidogenic proteins. Such strains typically vary in the basic underlying structures within the core region. In contrast, HTTex1 fibril types are interconvertible, being able to transition from P to T and then to N-fibrils in a process that can be reversed by treatments with organic solvent. This treatment rapidly reverts the CD and EPR spectra of the PRD from N- into T-fibrils, without significantly affecting the EPR spectra and solid-state NMR signals of the N17 and polyQ domains. Such a behavior is different from the various strains found in Aβ and α-synuclein where the architecture of the core region is faithfully propagated by seeding and the fibrils need to be completely disaggregated into monomers to transition from one polymorph to another. Thus, the different fibril polymorphs of HTTex1 do not exhibit the typical strain behavior seen in other amyloids.

HTTex1 consists of three separate domains of which the N17 and the polyQ are typically considered to promote aggregation[24,33], whereas the PRD is sometimes considered to inhibit aggregation[34,35]. Here we find that all domains partake in aggregation, but the time scales and structures are vastly different. Prior kinetic experiments demonstrated that the N17 initiates oligomerization on the minutes time scale and subsequently the polyQ regions engages in β-sheet formation on an hour time scale[24]. Here we find that the PRD also contributes to self-association as it promotes the coalescing of fibrils. This process is much slower than the N17 or polyQ-dependent aggregations, only proceeding on the order of days. Moreover, each of the domains uses a different secondary structure to promote aggregation. The N17 forms oligomers via helix–helix interactions, the polyQ forms fibrils via β-sheet structures, and the PRD leads to fibril–fibril interaction via the entanglement of PPII containing domains. The latter adhesive interactions are related to those formed by polyP peptides, which can form insoluble oligomers and films. All three domains and their interaction are of potential therapeutic interest. While blocking of the misfolding interactions of the N17 and polyQ region could be used to prevent fibril or oligomer formation altogether, it could be beneficial to promote the interactions between PRD region considering that the T-fibrils are more toxic than the more entangled N-fibrils.

Bäuerlein and co-workers detected unbundled and fibrils similar to our T- or P-fibrils in cellular inclusions using cryo-EM tomography[9]. Especially their data on C-terminally GFP-labeled HTTex1 confirmed our bottlebrush model by showing that the PRD that keeps the globular GFP at a distance from the fibril core. In addition, our observation that T-fibrils over time turn into less toxic N-fibrils is compatible with the observation of Ramdzan and co-workers that cellular HTTex1 inclusions became less dynamic over time and that this maturation correlated with a reduced rate of cell death[36].

Because the structural differences among the different fibril types are directly linked to toxicity, one would also expect that the PRD could be a contributor to HTTex1 toxicity. Intrabodies or proteins such as profilin, which bind to the PRD region have been shown to reduce HTTex1 aggregation and toxicity[37–41]. One of the factors that could contribute to the different toxicities of the HTTex1 fibril types is their seeding ability. Our in vitro studies show that the toxic T-fibrils are much more capable of acting as seeds than the nontoxic N-fibrils. This feature is consistent with the finding that the least seeding competent N-fibrils are also least toxic as shown by us and others[5]. Interestingly, the P-fibrils, which have the least degree of entanglement are the most potent seeds and are most toxic. Thus, seeding ability and toxicity seem inversely related with bundling of the fibrils. Mechanistically, this trend can be understood by the fact that both fibril ends, involved in primary seeding, and fibril surfaces, a source of potential secondary seeding, are less accessible in bundled fibrils. In addition, P- and T-fibrils are generally shorter than N-fibrils resulting in more fibril ends per monomer that can be involved in primary seeding. Similar seeding results were obtained using Neuro2a cells, where exogenously added fibrils were added to seed endogenously expressed HTTex1. Seeding ability has been considered an important factor in toxicity and cell-to-cell spreading of toxic species in a number of amyloid diseases. Our results therefore indicate that increased entanglement of the PRD protects from seeding and spreading of toxic species.

In addition to modulating seeding activity, the PRD region could further affect toxicity through protein–protein interactions. Huntingtin is a major hub for protein–protein interactions and the PRD plays a central role in many of these interactions because many huntingtin binding proteins contain SH3 domains and other PRD-specific binding regions. Fibril formation and oligomerization can have a major impact on the availability of the PRD for protein–protein interactions[23,41]. It has previously been proposed that the high density of surface-exposed bristles (such as those in P- and T-fibrils) can cause increased binding affinities for PRD binding proteins such as profilin[41]. We made similar observations with antibodies that bind to the PRD, which had a much stronger binding affinity to unbundled fibrils than to monomeric HTTex1[23]. Thus, fibrils or oligomers could further contribute to toxicity by having enhanced binding affinities to binding partners, which could cause mislocalization of HTTex1 binding partners. This notion is consistent with the fact that several PRD binding proteins have been shown to colocalize with cellular huntingtin aggregates[42–45]. Increased binding affinities to T-, P-fibrils, or oligomers could cause mislocalization and misregulation of such proteins. In contrast, the N-fibrils with their more tightly packed PRD regions are less likely to interact with such binding partners. In fact, we recently reported that PRD binding antibodies have a very weak affinity for the N-fibrils[23].

From a methodological perspective, the combined use of EPR distances, EPR mobilities, and solid-state NMR may well allow us to obtain a detailed three-dimensional structural model of the different fibril types, including detailed local structure for the N17 and polyQ region region. Such structural information should prove useful for the design molecules aimed at shifting the structural states of toxic species into nontoxic ones.

## Methods

**Protein expression and purification.** For the expression of unlabeled HTTex1, BL21(DE3) E. coli cells were first transformed with a pET32a-HDx46Q plasmid and plated onto LB agar with 100 mg/mL ampicillin (Amp)[7,11]. A single colony was added to 20 ml LB medium and incubated for about 3–4 h at 37 °C. This starter was then expanded to 1 l of LB/Amp and grown to an $OD_{600}$ of 0.7–0.8. Then, expression was induced by adding isopropyl 1-thio-β-D-galactopyranoside to a final concentration of 1 mM and cells were incubated overnight at 18 °C. Following expression, cell pellets were harvested by centrifugation (4500 g, 20 min, 4 °C). The expression of $^{13}C$, $^{15}N$-labeled HTTex1 was done similarly following the protocol by Marley and co-workers[46].

For the purification, cell pellets were resuspended in 20 mM Tris-HCl (pH 7.4), 300 mM NaCl, 10 mM imidazole containing 1% Triton X-100. Cells were lysed by incubation for 20 min at room temperature on a shaker followed by sonication using a model XL2000 ultrasonic cell disruptor (MICROSON) three times for 30 s with pulse mode output of 10 W. Cell lysate was clarified by centrifugation at 18,000 g for 20 min at 4 °C. The supernatant was then loaded on NiHis60 resin (Clontech) in a Econo-Pac column (Bio-Rad) and incubated for 1 h at 4 °C on a shaker. The column was then washed with several column volumes of 20 mM Tris-HCl (pH 7.4), 300 mM NaCl, 50 mM imidazole. The protein was eluted with 20 mM Tris-HCl (pH 7.4), 300 mM NaCl, and 300 mM imidazole. The eluted protein was diluted tenfold with 20 mM Tris pH 7.4 buffer and then further purified on a HiTrap Q XL column (GE Healthcare) with an AKTA FPLC system (Amersham Pharmacia Biotech) using a NaCl gradient (protein usually elutes between 150 and 350 mM NaCl).

Except for 86R1 (primers see Supplementary Table 1), all HTTex1 constructs containing single or double cysteine mutants were described previously[7,19,24]. For spin-labeling, HTTex1 constructs were expressed and purified as described above with all buffers prior to the elution from NiHis60 resin containing 1 mM dithiothreitol (DTT). Prior to elution, DTT was removed with 100 ml of additional

column wash containing no DTT. Protein was eluted with 25 ml of elution buffer (20 mM Tris, 300 mM NaCl, 250 mM imidazole, pH 7.4). $OD_{280}$ measurements were used to estimate protein concentration via an extinction coefficient $\varepsilon =$ 14,105 $M^{-1} cm^{-1}$. Consequently, the MTSL (1-oxyl-2,2,5,5 tetramethyl-Δ3-pyrroline-3-methylmethanethiosulfonate) spin label was added to the protein in a three- to fivefold molar excess at room temperature and incubated for an hour. Unbound spin label was removed by diluting the sample tenfold with 20 mM Tris, pH 7.4, and loading it onto a HiTrap Q XL anion exchange chromatography column (GE Healthcare) on an AKTA FPLC system (Amersham Pharmacia Biotech). The protein was eluted using a NaCl gradient as described above.

**Fibril formation**. HTTex1 T-fibrils for NMR and EPR measurements were made by starting with the Trx-HTTex1 fusion proteins at a concentration of 25 µM (632 µg/ml) in 20 mM Tris, 150 mM NaCl with 5% molar ratio of HTTex1(Q46) seeds added. Fibril started forming after adding 1 unit of EKMax per 280 µg of protein. The reaction mixture was left without agitation at 4 °C for 1 week to complete fibril formation. HTTex1 N-fibrils were prepared similarly but by incubating them at 37 °C instead of 4 °C.

N-fibrils were transformed into T-fibrils by pelleting the fibrils (60 min at 150,000 g) and redissolving the pellet in 0.5% TFA in $H_2O$. The fibrils could be further disaggregated into P-fibrils by adding an additional washing step with HFIP. Here the N-fibril pellet was redissolved in HFIP. The HFIP was consequently evaporated under a stream of $N_2$ and the dried fibril pellet was again redissolved in 0.5% TFA. P-fibrils can also be made from T-fibrils by pelleting the fibrils, redissolving them in 0.5% TFA, followed by sonication. P-fibrils prepared that way were also used as seeds to make T- and N-fibrils described above.

Fibrils were analyzed using EM for morphology and a BCA assay (ThermoFisher Scientific) was used to determine their concentration.

**Electron microscopy**. Grids for electron microscopy (150 mesh copper) were prepared by pipetting 10 µl of sample on them and letting the sample absorb for 5–10 min. For negative stain, 10 µl of 1% uranyl acetate solution was added and the grids were incubated for 2 min and then dried at room temperature for 1 h. All grids were captured on a Gatan digital camera as part of a JEOL JEM-1400 electron microscope (JEOL, Peabody, MA) at 100 kV using DigitalMicrograph 1.85.1535 and analyzed using ImageJ 1.52.

**Circular dichroism (CD)**. CD spectra of HTTex1 fibrils were measured at a concentration of 5–20 µM in 10 mM phosphate buffer with no salt, pH 7.4. All CD spectra were acquired on a Jasco 815 spectropolarimeter (Jasco, Inc., Easton, MD) using the Jasco Spectrum Measurement program. Data points were measured every 0.5 nm at a scan speed of 50 nm/min from 260 to 190 nm. Between 20 and 30, acquisitions were averaged for each sample. Background spectra of buffer only were subtracted to obtain the final spectrum. Temperature dependence of the CD spectra was measured using a Jasco PFD-425s temperature controller. PolyP solutions were incubated at a given target temperature for at least 20 min before recording the CD spectra. All experimental details stayed as described above. The difference spectra of N- and T-fibrils and polyP CD spectra recorded at 90 and 25 °C were smoothed using the Savitzky-Golay method in SciPy 1.1.0 in a python 3.7.3, NumPy 1.16.2 environment.

**EPR spectroscopy**. HTTex1 fibrils were sedimented using ultracentrifugation, resuspended in 20 µl of 20 mM Tris, 150 mM NaCl, and filled into quartz capillaries (ID = 0.6 mm and OD = 0.84 mm, VitroCom, Mt. Lakes, NJ). EPR spectra were recorded on an X-band Bruker EMX spectrometer (Bruker Biospin) at ambient temperature using WinEPR 4.33. The sweep width was 150 G using a HS cavity at an incident microwave power of 12.60 mW. All spectra were normalized by double integration.

Kinetic studies of fibril formation were done as described previously[24]. In short, a 15 µM solution of HTTex1 spin-labeled at residue 35 (35R1) was mixed with HTTex1(Q46) P-, T-, or N-fibril seeds at a 5% molar ratio or no seeds. The sample was then loaded into quartz capillaries and a EPR spectrum was recorded every 15 min for 17.5 h using the parameters described above.

To determine the distance between spin labels, four pulse DEER experiments[47,48] were measured at a temperature of 78 K on a Bruker Elexsys E580 X-band pulse EPR spectrometer equipped with a 3 mm split ring (MS-3) resonator, a continuous-flow cryostat (CF935, Oxford Instruments), and a temperature controller (ITC503S, Oxford Instruments). Trx-HTTex1 fusion protein with spin label pairs were diluted with 90–95% non-spin-labeled Trx-HTTex1 before inducing fibril formation with EKMax to reduce background from intermolecular distances. For some samples 15–20% sucrose was added as cryoprotectant after fibril formation. The fibril samples were flash-frozen and measured. The data were fitted using a single Gaussian as implemented in DEER Analysis 2019[49].

**Solid-state NMR**. All solid-state NMR experiments were acquired on a 600 MHz Agilent DD2 spectrometer using a 1.6 triple-resonance probe operating at 25 kHz MAS, 0 °C. Hard pulses nutation frequencies for $^1H$ and $^{13}C$ were 200 and

100 kHz, respectively. $^1H$-$^{13}C$ Hartmann-Hahn CP was done with using spin-lock nutation frequencies of 85 kHz for $^1H$ and 60 kHz for $^{13}C$ with a 10% ramp in $^1H$ radio-frequency amplitude. Two pulse phase modulation $^1H$ decoupling with an RF field strength of 120 kHz was used during indirect and direct acquisitions. A pre acquisition delay of 3 s was used for all NMR experiments. One dimensional $^{13}C$ CP and refocused INEPT (insensitive nuclei enhanced by polarization transfer) spectra were recorded with a spectral width of 50 kHz and 650 complex points. For the fibrils prepared at 4 and 37 °C, 256 scans were acquired. The 1D spectra of the protofibrils were acquired with 1024 scans. 2D DARR (dipolar assisted rotational resonance)[50] spectra were recorded with a 25 kHz $^1H$ spin-lock recoupling rf-field, a mixing time of 50 ms, and 50 kHz spectral widths in both dimensions with 650 complex points in the direct and 300 complex points in the indirect dimension. The DARR spectra of fibrils prepared a 4 °C, 37 °C, protofibril, were recorded with 16, 32, and 48 scans for each indirect increments, respectively. $^1H$-$^{13}C$ INEPT heteronuclear correlation (HETCOR) spectra were recorded with a $^{13}C$ refocusing pulse in the indirect $^1H$ dimension, which had a spectral width of 10 kHz, and was sampled with 16 scans for each of the 128 complex points. The direct $^{13}C$ dimension had a spectral width of 50 kHz and was sampled with 650 complex points. All spectra were referenced to 4,4-dimethyl-4-silapentane-1-sulfonic acid using adamantane as an external standard[51]. Spectra were recorded with VNMRJ 4.0, processed with nmrPipe 9.4[52], and plotted using nmrglue 0.9[53] based python scripts.

**Dot blot**. HTTex1 fibril and monomer samples (1.5 µl per sample) were blotted on a nitrocelluose membrane and incubated with the desired primary antibody at a 1:5000 dilution. The MW8 antibody was a gift from Ali Khoshnan but can also be purchased (Sigma-Aldrich Cat. No. MABN2529), the monoclonal Anti-polyHisitinde antibody was purchased from Sigma-Aldrich (Cat. No. OB05). A fluorescently-labeled secondary antibody (IRDye-800CW, Goat anti-Mouse IgG P/N 925-32210 Lot# C61012-06) was used at a dilution of 1:10,000. The dot blot was visualize using a LI-COR Odyssey Infrared Imaging system.

**Cellular puncta formation assay**. Neuro2a cells, grown on glass dishes coated with Matrigel, were transfected with plasmid constructs using a Lipofectamine LTX with Plus Reagent (ThermoFisher) transfection kit according to manufacturer protocol. The plasmid constructs containing HTTex1 with C-terminal EGFP and polyQ repeat lengths of Q16, Q39, Q72, or EGFP alone as control were described previously[25]. Co-transfection was performed in the same manner described by Nekooki-Machida et al. with the addition of 2.5 µg of fibrils 24 h after plating using Lipofectamine LTX. BSA fibrils were used as a control. To make BSA fibrils, lyophilized BSA was dissolved in PBS at a concentration of 100 µM, incubated for 1 h at 40 °C, sonicated, and stored in 4 °C until use. To confirm the intracellular delivery of protein in the co-transfection protocol, HTTex1 labeled with Alexa Fluor 594 at residue 101 was transfected and its location confirmed by fluorescence microscopy. Neuro2a culture were incubated for 48 h and imaged live using a Zeiss AxioPlan2 microscope. Fluorescent and differential interference contrast images were acquired using a 40× magnification. For each condition, the experiment was repeated three times and images of three visual fields were taken and analyzed for each repeat. The total number of transfected cells expressing EGFP and number of cells with EGFP foci of huntingtin aggregates was counted in a blinded fashion. Two-sided student T tests were used for statistical analysis to compare the various conditions vs. control.

**Cell toxicity assay**. The ST*HDH Q7/111* cell line (ID CH00096) was purchased from the Coriell Institute (Cat #: CH00096) and maintained following a protocol by Trettel et al.[26]. Cells were first grown in 20 ml complete culture media (80% Dulbecco's modified Eagle's medium; 10% fetal calf serum (FBS); 100 IU penicillin; 100 µg/ml streptomycin; 400 µg/ml G418) in a T75 flask (Genesee Scientific) at 33 °C. Then cells were treated with trypsin and neutralized with culture media. Twenty microliters of cell suspension were then taken from the flask and seeded into a 24-well plate. A cover glass was put into each well of these plates and coated with 10 mg/ml Poly D Lysine (Sigma-Aldrich) previous to adding the cells. After 24 h of incubation at 33 °C, N-, T-, and P-fibrils were added to a final protein concentration of 2 µM. We added a comparable volume of 0.025% (v/v) TFA buffer to our control. The culture media was changed 3 days after that and cells were fixed with 3.7% paraformaldehyde 5 days after fibril addition and stained with DAPI (1 µg/ml final concentration). After fixation and labeling, cells were imaged with LSM 800 microscope (Zeiss) and images were analyzed using ImageJ 1.52[54]. Cell counting was done blindly by another person. To determine the fractions of cells with nuclear fragmentation, the number of cells showing nuclear fragmentation and the total number of cells in ten representative fields were determined. The total number of cells per field for each of the conditions at the time of counting were similar, on the order of 800 cells (dead and live). This similarity in cell density suggests that any potential floating off of cells did not differentially affect the different groups.

**Reporting summary**. Further information on research design is available in the Nature Research Reporting Summary linked to this article.

## Data availability

Source data are provided with this paper. Other data are available from the corresponding authors upon reasonable request.

## Code availability

Processing scripts used to analyze and plot these data are available from the corresponding authors upon request.

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

## Acknowledgements

A.B.S. and R.L. would like to acknowledge funding from the National Institute of Neurological Disorders and Strok (R01NS084345) and the CHDI Foundation (Award A-12640).

## Author contributions

J.M.I. and N.K.P. prepared samples, measured and analyzed EPR, CD, and EM data. H.X. measured and analyzed cell toxicity data. K.T. and A.K.O. measured and analyzed cell seeding data. E.K.F. optimized cell toxicity assay. A.R., A.A., and F.M. acquired data and prepared fibril samples. J.C. coordinated and analyzed cell toxicity data. R.L. conceived the study, coordinated the EPR work and its interpretation, and co-wrote the manuscript. A.B.S. recorded and analyzed the NMR data, analyzed EPR data, made artwork and figures, and co-wrote the manuscript.

## Competing interests

The authors declare no competing interests.
