## [Peer Review File · Nature Communications]

Reviewers' comments:

Reviewer #1 (Remarks to the Author):

This study investigates the structural properties of two previously described fibril subtypes of Httex1 (N and T fibrils) that have differential toxicity to cell cultures using an array of biophysical methods, most notably NMR and EPR. The work concludes that the major difference between the fibril types is how they pack together via the polyproline region in the C-terminal end of Httex1. The more toxic fibrils are less well packed and it is predicted that toxicity emanates from polypro regions that are more exposed to interact with other cellular ligands. The authors were able to further dissociate fibrils into smaller units, which they call P fibrils. The smaller the fibril, the more able they were to nucleate aggregation in cell culture.

The study is well written and comprises of high-quality data that is in the most part robustly interpreted. It offers a very nice molecular explanation for the toxicities of Httex1 fibrils and model that is likely more broadly applicable to other amyloid assemblies.

Technical comments for this study:

1. Given that the study builds on the prior work (Nekooki-Machida et al., 2009) it is important that the authors are able to independently replicate the key findings from that work— notably that the T and N fibrils are differentially toxic to cell culture. By extension, I would recommend the authors investigate also the P fibrils, which might be anticipated to be the most toxic.
2. The authors conclude that the difference in the CD spectra between T and N fibrils arises from light scattering rather than differences in secondary structure and notably the light scattering disproportionately suppresses the CD signal at lower wavelengths. This point I think is very important and may not be obvious to those who don't work regularly with CD. I suggest the point be made more clearly at the first instance the data is presented (eg around line 70 to 84).
3. The argument that the EPR distance measurements reflect intramolecular distances is weakly justified in the experimental design. The conclusion might be more compelling if, for example, a dose curve was performed varying the ratios of the labelled:unlabelled Httex1.
4. The authors used TFA to dissolve the fibrils. Unclear what TFA is – do they mean trifluoroacetic acid? If so, it is unclear what the rationale of using that was. If they mean TFE – trifluoroethanol, then this would make more sense to me. Regardless, more details on these points are needed. On a related note, I presume HFIP is hexafluoroisopropanol?
5. In the cell seeding experiments, it is possible that trace amounts of toxic solvents (eg TFA or HFIP etc) induce the phenotypes observed. Additional controls are needed to account for these effects.
6. Also for the seeding experiments, the cells were examined for aggregates after 48 hours. However, it is well known in the literature that polyQ-expanded Httex1 is toxic and that cells lacking aggregates die more quickly than those with aggregates. It is also well known in the literature that aggregation is concentration dependent (ie cells expressing higher levels of Httex1 will aggregate faster than cells with lower levels). These effects could distort the conclusions about seeding if cells are preferentially lost due to toxicity or the different treatments have differing levels of expression. Additional controls to account for these effects are warranted.
7. The authors use anti-polyHisitinde antibody. Is this a typo for polyHistidine?
8. The conclusion that fibril packing is mediated by polypro helix is very interesting and is a testable hypothesis. It would be interesting to determine the importance of the polypro helix by mutagenesis of this region to disrupt that structure.

Reviewer #2 (Remarks to the Author):

In this manuscript Isas et al. examine the structural properties of three types of huntingtin (HTTex1) fibrils: toxic fibrils, nontoxic fibrils, and protofibrils. Using a range of biophysical techniques, including NMR spectroscopy, EPR, and CD, it is observed that a key difference between nontoxic and toxic fibrils lies in the C-terminal proline rich domain, which is more entangled in the nontoxic fibrils. The entanglement results in large, bundled/clumped fibrillary aggregates that exhibit decreased seeding

competency and reduced interactions with antibodies. It is speculated that these properties could explain the diminished toxicity of these fibrils. The authors show that toxic fibrils convert to nontoxic fibrils over time and that the process can be reversed using organic solvents. The core of all three types of fibrils, however is structurally similar. It is proposed that the fibril characteristics are determined by different C-terminal entanglement and not by different core structures as observed for fibril strains of other amyloids. This is a novel and potentially important concept that should be of interest to a broader readership. However, there are a few issues that need to be addressed.

1. The authors state that all fibril types can be reversibly interconverted. However, there is no experimental evidence provided that P-fibrils convert into N- or T-fibrils. Also, there is no discussion on whether the conversion of N- to T- or P-fibrils might be of biological relevance and if so what mechanisms might substitute the role of organic solvents used in this study.

2. It is stated that decreased entanglement enhances the seeding competency. There is no discussion provided that considers the number of fibril ends. It appears that seeding competency could be explained by the number of fibril ends, which may be largest for P-fibrils.

3. The INEPT spectra in Fig. 5 include T-fibrils after 12 months of incubation. The data is a little inconsistent when compared to the CD and EPR data which show T-fibrils after a few days of incubation. Would shorter incubation times also result in a disappearance of the His signature? Is there any chance that the C-terminus degraded?

4. It would be useful to combine the DEER data in the supplement with Fig. 3 in the main text. Also, the time scales of the dipolar evolution curves should be changed from “ns” to “ μ s”.

5. It might be helpful to specifically label the different domains in Fig. 2A (above the colored regions). In addition, the C-terminal His-tag (cyan) that is mentioned in the caption is not depicted.

6. Panel D is not properly listed in the caption of Fig. 6.

7. The statistical analysis in the manuscript is appropriate.

Reviewer #3 (Remarks to the Author):

In this manuscript, the authors tried to demonstrate that the toxicity of Huntingtin protein (HTTex1) amyloid fibers are correlated to the degree of entanglement of the C-terminal proline-rich domain (PRD), which does not form fibril core, for three types of fibrils (T, N, P-types). It has been known that the fiber core structures of the poly-Q region for T and N types of HTTex1 fibrils are likely similar. The authors tried to address the molecular-level origin of the differences for these fibril types by careful examination of HTTex1 fibers by NMR, CD, and EPR. Although the manuscript is enjoyable for specialists like the reviewer, my recommendation is that in the current form, the manuscript is more suited for publication in a specialized journal in chemistry/biophysics rather than Nature Communications. For example, the structural nature of the PRD interaction/entanglement is unclear. The entanglement of the PRD is merely indicated by the lack of signals of His-tag, which is not a part of the HTTex1. No detailed atomic-level interactions are demonstrated. At least, some key mechanism driving the interactions should be experimentally identified. It is also unclear how the degree of the entanglement for the three types of the fibrils can be correlated to the toxicity (no toxicity data). The story line is too complicated for most of non-specialists. The organization needs to be significantly improved to highlight the main points. As a result, the manuscript would not appeal to a broad range of audiences in the current form.

Response to Reviewers

We would like to thank the reviewers for their insightful comments and constructive suggestions. In the following, we answer their remarks in *italics*.

Reviewer #1:

This study investigates the structural properties of two previously described fibril subtypes of Httex1 (N and T fibrils) that have differential toxicity to cell cultures using an array of biophysical methods, most notably NMR and EPR. The work concludes that the major difference between the fibril types is how they pack together via the polyproline region in the C-terminal end of Httex1. The more toxic fibrils are less well packed and it is predicted that toxicity emanates from polypro regions that are more exposed to interact with other cellular ligands. The authors were able to further dissociate fibrils into smaller units, which they call P fibrils. The smaller the fibril, the more able they were to nucleate aggregation in cell culture.

The study is well written and comprises of high-quality data that is in the most part robustly interpreted. It offers a very nice molecular explanation for the toxicities of Httex1 fibrils and model that is likely more broadly applicable to other amyloid assemblies.

Technical comments for this study:

1. Given that the study builds on the prior work (Nekooki-Machida et al., 2009) it is important that the authors are able to independently replicate the key findings from that work– notably that the T and N fibrils are differentially toxic to cell culture. By extension, I would recommend the authors investigate also the P fibrils, which might be anticipated to be the most toxic.

We agree that having toxicity data would be an important. Following this suggestion, we tested multiple approaches to determine fibril toxicity and settled on nuclear fragmentation in STHdh Q7/111 cells as the most reliable method. As can be seen in the new Fig. 7e and f. T and to a larger degree P-fibrils lead to an increase in toxicity when added to cells whereas N-fibrils cause no statistical significant increase in toxicity over our control. These data are now included in the revised version of the manuscript.

2. The authors conclude that the difference in the CD spectra between T and N fibrils arises from light scattering rather than differences in secondary structure and notably the light scattering disproportionately suppresses the CD signal at lower wavelengths. This point I think is very important and may not be obvious to those who don't work regularly with CD. I suggest the point be made more clearly at the first instance the data is presented (eg around line 70 to 84).

Thank you for raising this point as it is important to point out that structural effects as well as scattering are likely responsible for the observed changes in the CD spectra. We think that the change in minima between the CD spectra of T and N fibril reflects some structural difference, because we see the corresponding shifts in minima with comparable difference spectra for PolyP film formation (see Figure 4). The polyP film is completely clear and does not show any signs of scatter, therefore, we do not think that scatter is solely responsible for the observed changes. However, work from many other

amyloid fibrils has shown that scattering can indeed decrease the CD intensities. Thus, it appears quite likely that scattering also contributes to the observed CD changes (see Figure S1). We now mention this in the revised version of the manuscript.

"In agreement with the original study, we found that the two different fibril types exhibited distinctively different CD-spectra (Fig. 1a). While T-fibrils typically had a minimum around 213 nm, that of N-fibrils was significantly shifted over to higher wavelengths (218-225 nm). These data are indicative of significant structural differences between the different fibril types. An additional, difference between the two fibril types was their optical appearance. While the T-fibrils were translucent, the N-fibrils were turbid, indicating that these fibrils were larger with a higher propensity to scatter light (Fig. 1b), which likely caused their lower relative CD intensity (Fig. S1)."

3. The argument that the EPR distance measurements reflect intramolecular distances is weakly justified in the experimental design. The conclusion might be more compelling if, for example, a dose curve was performed varying the ratios of the labelled:unlabelled Httex1.

This is a very good point that would have likely caused unnecessary confusion for many readers. Intermolecular distances can indeed become a concern in EPR DEER measurements. However, this issue has come up in numerous prior studies (e.g. Bedrood et al. 2012) and we have now developed standardize methods to overcome intermolecular distances by dilution. Here we made fibrils in which only 5-10% of the protein was labeled and 95-90% of the protein was unlabeled. This ratio was arrived at in optimization experiments in which fibrils from singly labeled proteins were generated and their DEER decay was analyzed. What one looks for is a slow decay of the DEER signal over time, which can then be subtracted out as background. A slow decay will not be possible, if short, intermolecular distances are present. Direct indication that this approach worked is presented in the data for the 63-102 labeling pair. Here, the decay of the DEER signal is extremely slow, meaning that the distance between spin-labels is so long that that it cannot be resolved by DEER. This would not be possible, if short intermolecular distances were present. The revised manuscript now explains this strategy more explicitly.

"Toward this end, we generated four different labeling pairs in the PRD. Two of the labeling pairs, 63-75 and 91-102, flank the two uninterrupted polyP repeats P11 and P10, respectively. The 75-91 pair borders a Pro-rich linker region L17, which connects the two polyP repeats. The 63-102 labeling pair was designed to obtain distance information encompassing the entire PRD (Fig. 3a). We sparsely incorporated these derivatives (10-5%) into N or T-fibrils grown from excess (90-95%) unlabeled protein. This dilution of spin labeled monomers with a majority of unlabeled protein avoids the influence of intermolecular distances in our measurements, which is further removed via background subtraction. The very slow decay of the DEER signal for the 63-102 labeling pair (Fig. S2) indicates the absence of short intermolecular distances and supports the success of this dilution approach."

4. The authors used TFA to dissolve the fibrils. Unclear what TFA is – do they mean trifluoroacetic acid? If so, it is unclear what the rationale of using that was. If they mean TFE – trifluoroethanol, then

this would make more sense to me. Regardless, more details on these points are needed. On a related note, I presume HFIP is hexafluoroisopropanol?

The revised manuscript now defines both abbreviations. We indeed use trifluoroacetic acid, which we found best suited to break up the polyP film after trying several other approaches. HFIP is indeed hexafluoroisopropanol.

5. In the cell seeding experiments, it is possible that trace amounts of toxic solvents (eg TFA or HFIP etc) induce the phenotypes observed. Additional controls are needed to account for these effects.

This is an important point, especially since HFIP can disrupt membranes and be toxic. However, HFIP was only necessary to disaggregate N-fibrils into P-fibrils. Fresh P-fibrils can be made using small amounts of TFA and sonication. Added an EM of these fibrils to the supplement (Fig. S4) to clarify this point. N-fibrils and T-fibrils are made without TFA. We have now included new data to show that TFA treatment is not responsible for cell toxicity as the buffer control in Figure 7E and 7F contains the same trace amount of TFA as the P-fibril sample.

6. Also for the seeding experiments, the cells were examined for aggregates after 48 hours. However, it is well known in the literature that polyQ-expanded Httex1 is toxic and that cells lacking aggregates die more quickly than those with aggregates. It is also well known in the literature that aggregation is concentration dependent (ie cells expressing higher levels of Httex1 will aggregate faster than cells with lower levels). These effects could distort the conclusions about seeding if cells are preferentially lost due to toxicity or the different treatments have differing levels of expression. Additional controls to account for these effects are warranted.

Thank you for this comment. While generating the cell toxicity data for this manuscript, we found that HTTex1 cell toxicity only occurs in certain cell types, after several days of incubation. Our cell seeding experiments were done at an earlier time point and in a less sensitive cell line compared to these toxicity assays. The revised version of the manuscript now shows these toxicity data (Figure 7E and F) indicating that T and P-fibrils are toxic whereas N-fibrils have little or no toxicity. The correlation of the ability to induce intracellular HTTex1 aggregation with fibril toxicity indicates that the fibrils are more toxic than soluble, polyQ expanded HTTex1. In addition, our cell seeding data show the fraction of cells with puncta relative to all cells that have been transfected. The HTTex1 constructs used in these experiments showed no significant variation of transfected cells per frame. We now mentioned this in the revised version of the manuscript.

"As controls, we also used cells expressing HTTex1-EGFP with a short Q-length (Q16), which are known to have a low fibril forming propensity or EGFP alone. Neither of these cells exhibited any detectable change in puncta from treatment with the different fibrils indicating that seeding effect, which was quantified by the relative puncta formation of all transfected cells, is specific to HTTex1 with expanded Q-lengths. We did not observe any significant change in the number of transfected cells independent of Q-length and seeding."

7. The authors use anti-polyHisitinde antibody. Is this a typo for polyHistidine?

Anti-polyHisitinde is the name under which the vendor (Sigma-Aldrich) sells this antibody.

8. The conclusion that fibril packing is mediated by polypro helix is very interesting and is a testable hypothesis. It would be interesting to determine the importance of the polypro helix by mutagenesis of this region to disrupt that structure.

Our work already tests this hypothesis from multiple angles. I) we showed using EPR that there is a change in the PRD with bundling but no change to the other domains. II) our solid-state NMR data show a decrease in dynamics of the PRD between T-fibrils and N-fibrils whereas the structure of the polyQ fibril core is the same for both fibrils. III) our CD spectra show the structural difference between bundled and unbundled fibrils is analogous to a polyPro film formation. IV) An antibody with an epitope in the PRD (MW8) preferentially binds disaggregated fibrils. We now make these points more clearly in the revised manuscript.

“T-fibrils and P-fibrils exhibited the characteristic bottle brush structures where bristles have high mobility and radiate outward. In contrast, the bristles in N-fibrils are less dynamic and coalesced resulting in highly bundled fibrils (Fig.8). That the bundling of N-fibrils is caused by PRD aggregation in a mechanism analogous to polyPro film formation is supported by the following experimental findings. I) EPR data showing only a change in the PRD with bundling. II) Solid-state NMR data showing a less dynamic PRD in N compared to the T and P-fibrils. III) CD spectra showing that the structural difference between bundled and unbundled fibrils is analogous to polyPro film formation. IV) A dot blot showing that the antibody MW8 which recognizes an epitope in the PRD preferentially binds T and P-fibrils. In addition, our EPR, solid-state NMR, and CD data supports the notion that N-fibrils can form from T-fibrils via the coalescence of the PRD bristles. Our model is furthermore consistent with the greater diameter of N-fibrils and the fact that all fibril types can be interconverted”

Our previous work (Isas et al. 2015) also showed that unbundled fibrils are separated by the length of two C-termini in a PPII conformation indicating that this domain forms fibril surface.

Regarding mutagenesis experiments, there is no guarantee that those won't create a new surface that will allow bundling. For example, Crick and co-workers showed that truncating the PRD increases the ability of HTTex1 fibrils to bundle (Crick et al. PNAS 2013). We would interpret their data in terms of the polyQ domain having formed a new sticky surface, as we also find that fibrils from polyQ peptides are very bundled.

Reviewer #2:

In this manuscript Isas et al. examine the structural properties of three types of huntingtin (HTTex1) fibrils: toxic fibrils, nontoxic fibrils, and protofibrils. Using a range of biophysical techniques, including NMR spectroscopy, EPR, and CD, it is observed that a key difference between nontoxic and toxic fibrils lies in the C-terminal proline rich domain, which is more entangled in the nontoxic fibrils. The entanglement results in large, bundled/clumped fibrillary aggregates that exhibit decreased seeding competency and reduced interactions with antibodies. It is speculated that these properties could explain the diminished toxicity of these fibrils. The authors show that toxic fibrils convert to nontoxic fibrils over time and that the process can be reversed using organic solvents. The core of all three types of fibrils, however is structurally similar. It is proposed that the fibril characteristics are determined by different C-terminal entanglement and not by different core

structures as observed for fibril strains of other amyloids. This is a novel and potentially important concept that should be of interest to a broader readership. However, there are a few issues that need to be addressed.

1. The authors state that all fibril types can be reversibly interconverted. However, there is no experimental evidence provided that P-fibrils convert into N- or T-fibrils. Also, there is no discussion on whether the conversion of N- to T- or P-fibrils might be of biological relevance and if so what mechanisms might substitute the role of organic solvents used in this study.

Thank you for this comment. It is correct that our work does not imply the reversibility of fibril polymorphs inside the cell. We rather illustrate the special nature of HTTex1 polymorphs that is not due to a structural change in the fibril core unlike other amyloid strains. Here we correlate structure and toxicity and it will be up to further studies to confirm these mechanism in vivo. That being said, T-fibrils have been observed in cellular inclusions (Bäuerlein et al. 2017) and cellular HTTex1 inclusions have been described to become less dynamic over time with consequences for cell fate (Ramdzan et al. 2017). We now discuss this in the revised version of the manuscript.

"Bäuerlein and co-workers detected unbundled and fibrils similar to our T or P-fibrils in cellular inclusions using cryo-EM tomography (Bauerlein et al., 2017). Especially their data on C-terminally GFP labeled HTTex1 confirmed our bottlebrush model by showing that the PRD that keeps the globular GFP at a distance from the fibril core. In addition, our observation that T-fibrils over time turn into less toxic N-fibrils is compatible with the observation of Ramdzan and co-workers that cellular HTTex1 inclusions became less dynamic over time and that this maturation correlated with a reduced rate of cell death (Ramdzan et al., 2017)."

2. It is stated that decreased entanglement enhances the seeding competency. There is no discussion provided that considers the number of fibril ends. It appears that seeding competency could be explained by the number of fibril ends, which may be largest for P-fibrils.

Thank you for highlighting this omission. We now discuss the question of fibril ends in our discussion section.

"Mechanistically, this trend can be understood by the fact that both fibril ends, involved in primary seeding, and fibril surfaces, a source of potential secondary seeding, are less accessible in bundled fibrils. In addition, P and T-fibrils are generally shorter than N-fibrils resulting in more fibril ends per monomer that can be involved in primary seeding."

3. The INEPT spectra in Fig. 5 include T-fibrils after 12 months of incubation. The data is a little inconsistent when compared to the CD and EPR data which show T-fibrils after a few days of incubation. Would shorter incubation times also result in a disappearance of the His signature? Is there any chance that the C-terminus degraded?

It is correct that a 12-month incubation time for the NMR data is significantly longer than the comparable EPR and CD data. Nevertheless, what these data show is the fact that T to N-fibril transitions can also be observed with NMR. The NMR data are inconsistent with the degradation of the C-terminus because all resonances of the 12 month spectrum in Figure 5c come from the C-terminus.

In addition, a hydrolysis of the C-terminal HIS-tag would result in an increased His signal in these type of spectra rather than a decrease.

4. It would be useful to combine the DEER data in the supplement with Fig. 3 in the main text. Also, the time scales of the dipolar evolution curves should be changed from “ns” to “μs”.

This is a good suggestion. However, the space constraint of the journal made us decide to put them into the supplement. Thanks for catching this typo. We changed the scale from ns to μs as suggested.

5. It might be helpful to specifically label the different domains in Fig. 2A (above the colored regions). In addition, the C-terminal His-tag (cyan) that is mentioned in the caption is not depicted.

Thank you for pointing this out. We changed the figure accordingly.

6. Panel D is not properly listed in the caption of Fig. 6.

Panel E was listed twice, we corrected this in the revised manuscript.

7. The statistical analysis in the manuscript is appropriate.

Thank you.

Reviewer #3 (Remarks to the Author):

In this manuscript, the authors tried to demonstrate that the toxicity of Huntingtin protein (HTTex1) amyloid fibers are correlated to the degree of entanglement of the C-terminal proline-rich domain (PRD), which does not form fibril core, for three types of fibrils (T, N, P-types). It has been known that the fiber core structures of the poly-Q region for T and N types of HTTex1 fibrils are likely similar. The authors tried to address the molecular-level origin of the differences for these fibril types by careful examination of HTTex1 fibers by NMR, CD, and EPR. Although the manuscript is enjoyable for specialists like the reviewer, my recommendation is that in the current form, the manuscript is more suited for publication in a specialized journal in chemistry/biophysics rather than Nature Communications. For example, the structural nature of the PRD interaction/entanglement is unclear. The entanglement of the PRD is merely indicated by the lack of signals of His-tag, which is not a part of the HTTex1.

We cordially disagree with the statement that the entanglement of the PRD is only shown by NMR. In fact, our manuscript presents evidence of this entanglement via I) EPR spectroscopy (decrease in relative EPR amplitude of spin labels in the PRD of N versus T and P-fibrils), II) NMR spectroscopy, III) CD spectroscopy (change in CD spectra between N and T-fibrils compatible with polyP film formation), IV) antibody binding (decreased accessibility of an antibody specific to the PRD in N compared to T-fibrils). We now list these conclusions in the first paragraph of the discussion section to make this point more clearly.

No detailed atomic-level interactions are demonstrated.

This is correct. However, because the PRD is an intrinsically disordered, low complexity sequence. Therefore, we do not expect that the interactions driving the bundling to have a well-defined interface that would permit an atomic-level description.

At least, some key mechanism driving the interactions should be experimentally identified.

We identified polyP film formation as the mechanism of interaction using CD spectroscopy. We feel that reproducing the fibril behavior with a minimal system (i.e. polyP) strongly supports the proposed mechanism. Further support for this mechanism comes from our EPR, NMR, and antibody binding data as described above.

It is also unclear how the degree of the entanglement for the three types of the fibrils can be correlated to the toxicity (no toxicity data).

The revised manuscript now includes toxicity data that confirm that N-fibrils have little to no toxicity whereas T and P-fibrils are toxic.

The story line is too complicated for most of non-specialists. The organization needs to be significantly improved to highlight the main points. As a result, the manuscript would not appeal to a broad range of audiences in the current form.

Our article presents a lot of data from different techniques and we worked hard to present them as comprehensively as possible. Reviewer 1 above agreed that we succeeded to do so, but the same reviewer also pointed out the need to define some of the abbreviations. We have now defined those abbreviations, which will help to remove some ambiguity. Moreover, to further highlight the main points of the manuscript and make it even less complicated to read for the non-specialist, we now include subheadings in the Results section. We also realized that the comprehensive line of evidence in support of the polyP dependent fibril bundling was never properly summarized. This is now included. We feel that these changes will make it easier for the reader to readily appreciate the points made in this study.

Reviewers' comments:

Reviewer #1 (Remarks to the Author):

I thank the authors for their careful consideration of my comments and suggestions and for addressing them. I have one outstanding concern regarding the new data interrogating the toxicity of the fibrils (point 1 of the response to authors). I appreciate the authors taking the time to independently validate the previously reported toxicity of the different fibrils. However, I have two concerns with the assay that was used and whether it is appropriate for producing reliable data. The first centers on the problem that microscopic analysis can be very subjective and biased to particular areas on the plate that was imaged. This subjectivity can be readily addressed if the assay was performed blind (but this was not clear in the manuscript). The second issue is that measuring cell death by DAPI reactivity after several days incubation is highly unlikely to properly capture rates or extents of death in a quantitative manner. Once cells die, they dissociate from the culture plate and disintegrate - hence detection of DAPI puncta on the plate likely only captures a small fraction of the death and may not reflect the bulk population of cells. One approach to address this issue is to measure survival longitudinally - ie by tracking individual cells over time - or if the equipment to do that experiment is not available, then estimates of the cell density over time could work as a proxy for death (eg such as the MTT reduction assay and by directly measuring cell counts and death rates of all cells harvested from the plate and media at different time points).

Response second Review

We would like to thank the reviewer for the comment. In the following, we answer it in *italics*.

Reviewer #1:

I thank the authors for their careful consideration of my comments and suggestions and for addressing them. I have one outstanding concern regarding the new data interrogating the toxicity of the fibrils (point 1 of the response to authors). I appreciate the authors taking the time to independently validate the previously reported toxicity of the different fibrils. However, I have two concerns with the assay that was used and whether it is appropriate for producing reliable data. The first centers on the problem that microscopic analysis can be very subjective and biased to particular areas on the plate that was imaged. This subjectivity can be readily addressed if the assay was performed blind (but this was not clear in the manuscript). The second issue is that measuring cell death by DAPI reactivity after several days incubation is highly unlikely to properly capture rates or extents of death in a quantitative manner. Once cells die, they dissociate from the culture plate and disintegrate - hence detection of DAPI puncta on the plate likely only captures a small fraction of the death and may not reflect the bulk population of cells. One approach to address this issue is to measure survival longitudinally - ie by tracking individual cells over time - or if the equipment to do that experiment is not available, then estimates of the cell density over time could work as a proxy for death (eg such as the MTT reduction assay and by directly measuring cell counts and death rates of all cells harvested from the plate and media at different time points).

The cell counting was indeed done blind by another person who did not know the conditions. This is now mentioned in the methods section.

We agree with the reviewer that cells floating off could cause us to underestimate some dead cells. We also agree with the reviewer that time lapse live cell imaging experiments would have been highly useful. However, the core facility for live-cell imaging is inaccessible to us during the pandemic. We did, however, quantify the cell density in our experiments. As now mentioned in the manuscript, we found that the cell density in all experiments was highly comparable suggesting that any potential floating off of cells did not differentially affect the different groups. Thus, together with a previous extensive study which reported the same differential toxicity, we find that our interpretation of the toxicity data is on strong footing.